# XCO$_2$ observations using satellite measurements with moderate spectral resolution: Investigation using GOSAT and OCO-2 measurements

Lianghai Wu[1], Joost aan de Brugh[1], Yasjka Meijer[2], Bernd Sierk[2], Otto Hasekamp[1], Andre Butz[3], and Jochen Landgraf[1]

[1]SRON Netherlands Institute for Space Research, Sorbonnelaan 2, 3584 CA Utrecht, The Netherlands
[2]European Space Agency, Keplerlaan 1, 2201 AZ Noordwijk, The Netherlands
[3]University Heidelerg,Institut für Umweltphysik, Im Neuenheimer Feld 229, 69120 Heidelberg,Germany

**Abstract.**

In light of the proposed space segment of Europe's future CO$_2$ monitoring system, we investigate the spectral resolution of the CO$_2$ spectrometer, which measures Earthshine radiance in the three relevant spectral bands at 0.76, 1.61 and 2.06 $\mu$m. The Orbiting Carbon Observatory-2 (OCO-2) mission covers these bands with fine spectral resolution but limited spatial coverage, which hampers the monitoring of localized anthropogenic CO$_2$ emission. The future European CO$_2$ monitoring constellation, currently undergoing feasibility studies at the European Space Agency (ESA), is targeting a moderate spectral resolution of 0.1, 0.3 and 0.3-0.55 nm in the three spectral bands with high signal-to-noise (SNR) ratio as well as a spatial resolution of 4 km$^2$ and a across-track swath width $> 250$ km. This spectral and radiometric sizing is deemed to be favorable for large-swath imaging of point-sources of CO$_2$ emission. To assess this choice, we use real and synthetic OCO-2 satellite observations, which we spectrally degrade to the envisaged lower spectral resolution. We evaluate the corresponding CO$_2$ retrieval accuracy by taking the Total Carbon Column Observing Network (TCCON) observations as reference. Here, a lower spectral resolution enhances the scatter error of the retrieved CO$_2$ column mixing ratio (XCO$_2$) but has little effect on the station-to-station variation of the biases. We show that the scatter error gradually increases when decreasing spectral resolution. Part of the scatter error increase can be attributed to the retrieval noise error which can be compensated by a future instrument with improved SNR. Moreover, we consider the effect of the reduced spectral resolution on the capability to capture regional XCO$_2$ variations and XCO$_2$ plumes from selected OCO-2 orbits. The investigation using measurements from the Greenhouse gases Observing SATellite (GOSAT) and synthetic measurements confirms our finding and indicates that one major source of uncertainties regarding CO$_2$ retrieval is the insufficient information on aerosol properties that can be inferred from the observations. We hence recommend the implementation of simultaneous, co-located measurements that have larger information content on aerosols with an auxiliary instrument in the future European observing system.

# 1 Introduction

The atmospheric concentration of the most important anthropogenic greenhouse gas carbon dioxide ($CO_2$), is increasing rapidly due to fossil fuel combustion and changes in land use with serious environmental consequences such as global temperature rise, ocean acidification and an increase in extreme weather events (Cox et al., 2000; Caldeira and Wickett, 2003). At the same time, our knowledge about sources and sinks of $CO_2$ is still limited. Here, satellite observations of the column-averaged dry-air mole fraction of $CO_2$ ($XCO_2$) gives both scientists and policy-makers a powerful tool to develop and evaluate mitigation strategy in the face of future climate change. To derive $CO_2$ hot spot emissions and the strength of regional $CO_2$ sources and sinks, $XCO_2$ satellite observations are needed with unprecedented precision and accuracy, good spatial coverage, and high spatial resolution. For anthropogenic $CO_2$ monitoring, Ciais et al. (2015) and Crisp et al. (2018) listed the main driving requirements as a $XCO_2$ precision $\leq 0.7$ ppm and systematic error $\leq 0.5$ ppm with a spatial resolution of 4 km$^2$ and a swath of $> 250$ km with a coverage requirement of 2-3 days. Here the high accuracy and precision are needed because even the largest $CO_2$ surface sources and sinks produce only small changes in the atmospheric $XCO_2$.

The SCanning Imaging Absorption spectroMeter for Atmospheric CHartographY (SCIAMACHY) on-board ENVISAT (March 2002-April 2012) is the pioneering passive remote sensing spectrometer which can measure atmosphere $CO_2$ and $CH_4$ columns down to the Earth surface (Buchwitz et al., 2005). Currently, the Greenhouse Gases Observing Satellite (GOSAT, Yokota et al. (2009); Kuze et al. (2016)) and the Orbiting Carbon Observatory-2 (OCO-2, Crisp et al. (2017)) missions are in orbit, dedicated to observing $XCO_2$ from space. Additionally, the Carbon Monitoring Satellite (CarbonSat, Bovensmann et al. (2010); Buchwitz et al. (2013)) was proposed to the European Space Agency (ESA) with the objective to advance our knowledge on the natural and man-made sources and sinks of $CO_2$ from regional and country down to local scales, but was not selected for mission implementation. Recently, NASA's OCO-3 instrument was launched and mounted successfully on the Japanese Experiment Module-Exposed Facility on board the International Space Station (Eldering et al., 2019). As a successor of this series of dedicated greenhouse gas mission, CNES aspire to launch the MicroCarb satellite in the year 2021 (Pascal et al., 2017). Table 1 includes the spectral and spatial properties of GOSAT, OCO-2 and CarbonSat satellite instruments, observing the Earth-reflected sunlight in the oxygen ($O_2$) A-band around 0.765 $\mu$m, the weak $CO_2$ absorption band around 1.61 $\mu$m and the strong $CO_2$ absorption band around 2.06 $\mu$m. Among those instruments, the CarbonSat concept has the largest swath with good spatial resolution but with significantly reduced spectral resolution compared to GOSAT and OCO-2. At the same time, the CarbonSat sizing concept would offer a much higher signal-to-noise ratio and broader spectral bandwidth. These properties were chosen to enable simultaneous measurement of $CH_4$ in the 1.61 $\mu$m band (1.590-1.675$\mu$m), and include an additional $CO_2$ band (1.990-2.035 $\mu$m). The selected moderate spectral resolution is expected to reduce the sensitivity to instrument errors, e.g. distortions of the instrument spectral response function (ISRF) and detector non-linearity. It also enables the use of low-order diffraction grating technologies with high efficiencies and low straylight (Sierk et al., 2016). On the other hand the design introduces the risk of $XCO_2$ errors due to spectral interference with other absorbers and enhanced aerosol induced errors. To evaluate this risk Galli et al. (2014) analyzed a spectral degradation of GOSAT observations and the induced error on $XCO_2$.

Proceeding from the CarbonSat proposal and the Paris Agreement, which was signed in 2015 by 195 countries agreeing to combat climate change and to accelerate and intensify the actions for a sustainable low carbon future, the European Commission gave ESA the mandate to investigate the implementation of a satellite mission monitoring anthropogenic $CO_2$ emissions. To meet the mission objectives, a careful trade-off has to be made between different requirements. With the successful launch of

OCO-2 and the application of several algorithms to infer $XCO_2$ from the observations (Boesch et al., 2011; O'Dell et al., 2012; Wu et al., 2018), we have, next to GOSAT, an additional data set at hand to verify the impact of a reduced spectral resolution on $XCO_2$ retrieval. In particular, applying the same degradation approach to both GOSAT and OCO-2 observations may help to identify instrument specific aspects of the induced errors due to a reduced spectral resolution.

In this study, we investigated the retrieval performance of OCO-2 observations degraded to different spectral resolutions

building upon the work by Galli et al. (2014). We evaluate the $XCO_2$ retrieval accuracy and precision using both OCO-2 measurements and produce spectra with the reduced spectral resolution and the sampling ratio as listed in Table 1, which in the remainder of the study will be referred to as the moderate spectral resolution (MSR) concepts. Due to the coarser spectral resolution and sampling for the MSR concepts, the SNR performances enhances in the corresponding spectral bands. We first investigate the impact of reduced spectral resolution with simulated OCO-2 and MSR type measurements for a global

ensemble. For satelite observations, the differences between retrieved $XCO_2$ and collocated ground based observations from the Total Carbon Column Observing Network (TCCON) are used to estimate the retrieval uncertainty. We also compare $XCO_2$ retrievals over Europe, the Middle East and Africa (EMEA) regions and selected orbits with hot-spots as reported by Nassar et al. (2017) using both OCO-2 and MSR type measurements. A correspsonding analysis is done for GOSAT observations to relate our analysis to the previous work done by Galli et al. (2014).

The paper is organized as follows: Section 2 describes our approach to lower the spectral resolution of observed OCO-2 or GOSAT spectra and introduces the $XCO_2$ retrieval algorithm RemoTeC and its particular settings for this study. Section 3 summarizes the satellite observations and validation data used in the study and Section 4 evaluates the OCO-2 and GOSAT $XCO_2$ retrievals for the original and reduced spectral resolutions using collocated TCCON data. Here, the impact of a reduced spectral resolutions on $XCO_2$ retrievals is further investigated over EMEA regions and selected OCO-2 orbits with hot-spots.

Finally, section 5 concludes the paper with recommendations for a future European $CO_2$ monitoring mission.

## 2 Method

### 2.1 Retrieval method and setting up

To retrieve $CO_2$ columns from space-borne Earth-shine radiance observations in the 0.76, 1.61 and 2.06 $\mu$m spectral ranges with different spectral resolutions, we use the RemoTeC full-physics retrieval algorithm (Hasekamp and Butz, 2008), which was

first applied for GOSAT measurements and later extensively used for greenhouse gas retrievals of different missions including GOSAT, OCO-2 and Sentinel-5P (Butz et al., 2009; Schepers et al., 2012; Guerlet et al., 2013; Hu et al., 2016; Wu et al., 2018; Hu et al., 2018). The algorithm employs an iterative inverse scheme combined with an efficient forward radiative transfer model developed by Landgraf et al. (2001); Hasekamp and Landgraf (2005); Hasekamp and Butz (2008); Schepers et al. (2014). For

a given model atmosphere, the forward model simulates the intensity vector field, including its Stokes parameter Q and U on
a line-by-line spectral sampling, and its derivatives with respect to both the amount of all relevant trace gases and the optical
properties of spherical aerosols in different layers of the model atmosphere. Moreover, RemoTeC infers state parameters of
the atmosphere by minimizing the difference between forward model and satellite observations. Due to the different spectral
coverage of the 1.61 $\mu$m band and corresponding sensitivities, for GOSAT measurements 12-layer profiles of $CO_2$ and $CH_4$
partial column are retrieved whereas for OCO-2 measurements we only infer the corresponding $CO_2$ profile. Apart from that,
the algorithm setup is the same for both missions, which infers additionally: $H_2O$ total column, surface properties, spectral
shifts, intensity offsets and aerosol optical properties. To describe the size distribution of the atmospheric aerosol, RemoTeC
uses a power-law size distribution ($n(r) \propto r^{-\alpha}$ with the particle radius $r$ and retrieves the size parameter $\alpha$ and total amount of
aerosol particles $N$. Here, the size parameter $\alpha$ is unitless. For the aerosol height distribution, we assume a Gaussian profile with
a full-width-half-maximum of 2 km and retrieve its center height $h_{aer}$. For this study, we consider only satellite observations
over land, where we assume a Lambertian surface reflection model with describing the inter-band spectral dependence of the
surface albedo as a second order polynomial.

In terms of spectral calibration, we adjust spectral shifts for both the Earth radiance measurement and solar reference model
in each spectral band while an intensity offset is only fitted in the 0.76 $\mu$m band for both GOSAT and OCO-2 spectra. These
RemoTeC retrieval settings were also used in GOSAT retrievals by Butz et al. (2011); Schepers et al. (2012); Guerlet et al.
(2013); Buchwitz et al. (2017). It should be noted that in the recent study by Wu et al. (2018) we found that retrieving an
intensity offset in all three OCO-2 bands significantly improves the accuracy of the data product. Measurements of the OCO-2
push-bloom spectrometer with high SNR includes most likely larger stray light errors than the TANSO-FTS (Thermal And
Near infrared Sensor for carbon Observation - Fourier Transform Spectrometer) on-board GOSAT. In this study, however, we
use the same retrieval settings for both GOSAT and OCO-2 data for the following reasons:

1. A consistent retrieval setting can help to identify the origin of the product uncertainties. Assuming that the error analysis
   differs significantly for two satellite missions, it seems likely to be an instrument specific issue rather than due to the
   algorithm itself;

2. It turns out to be difficult to fit an intensity offset in the 2.06 $\mu$m band for spectra with a coarse spectral resolution of
   0.55 nm;

3. The primary target of the study is to understand the impact of a reduced spectral resolution and so the relative change of
   retrieval performances with spectral resolution is the main focus of this study.

To account for line mixing as well as collision-induced absorption of $O_2$ and $CO_2$ we employ the spectroscopic model
by Tran and Hartmann (2008). The molecular absorption database HITRAN 2008 is used for $CH_4$ and $H_2O$ considering
the Voigt line shape model. The algorithm also requires auxiliary information on vertical profiles of pressure, temperature
and humidity, and surface wind speed, which are adapted from the European Centre for Medium Range Weather Forecasts
(ECMWF). Surface elevation information is taken from the 90-meter digital elevation data of NASA's Shuttle Radar Topogra-
phy Mission (Farr et al., 2007). Prior information on $CO_2$ and $CH_4$ profiles are interpolated from CarbonTracker and the TM5

model for the years 2013 and 2010 (Peters et al., 2007; Houweling et al., 2014), while prior information of the surface albedo is estimated from the mean radiance of the observation. Aerosol priors are the same for all retrievals.

Cloud-contaminated observations are rejected by strict data filtering using prior non-scattering retrievals (Schepers et al., 2012) and so clouds do not need to be considered in the retrieval algorithm. Here, the cloud clearing relies on the fact that the difference of $CO_2$ and $H_2O$ columns, retrieved independently from the 1.61 and 2.06 $\mu$m bands for a non-scattering model atmosphere, indicates the measurement contamination by clouds (Taylor et al., 2016). Furthermore, the difference between the $O_2$ column inferred from the $O_2$ A-band with a non-scattering atmosphere and the corresponding column derived from

the ECMWF surface pressure can be used for cloud filtering. Additionally, we reject spectra with low signal-to-noise ratio, extreme viewing geometry, cirrus contamination and high aerosol load to avoid large retrieval errors. The applied quality filtering variables and corresponding ranges are listed in Table 2. The data screening is described in more detail by Detmers and Hasekamp (2015) and Wu et al. (2018) for the GOSAT and OCO-2 retrievals, respectively, where for OCO-2 the data screening does not rely on the intensity offsets in the 1.61 and 2.06 $\mu$m bands because it is not retrieved from the measurement

in the context of this study.

## 2.2    Degradation of spectral resolution

To simulate a spectral measurement $\mathbf{I}_{obs}$ from a top-of-atmosphere line-by-line spectrum $\mathbf{I}_{rad}$ we apply the convolution

$$\mathbf{I}_{obs}(\lambda_i) = (\mathbf{H}_i * \mathbf{I}_{rad})(\lambda_i) \tag{1}$$

$$= \int d\lambda \mathbf{H}_i(\lambda_i - \lambda)\mathbf{I}_{rad}(\lambda) \tag{2}$$

where $\mathbf{H}_i(\lambda_i - \lambda)$ is the instrument spectral response function (ISRF) of a spectrometer at central wavelength $\lambda_i$. The spectral resolution of the spectrometer is characterized by the full width at half maximum (FWHM) of the ISRF. This equation holds both for the spectra recorded by OCO-2 and GOSAT and for the spectra degraded in spectral resolution but with different ISRFs. To estimate the ISRF $\mathbf{H}_i^{\mathrm{deg}}$ of a degraded spectral resolution, we convolve the original GOSAT and OCO-2 ISRF $\mathbf{H}_i$ with a Gaussian function $\mathbf{g}$,

$$\mathbf{H}_i^{\mathrm{deg}} = \mathbf{H}_i * \mathbf{g} \tag{3}$$

with

$$\mathbf{g} = Ae^{-\frac{(\lambda-\lambda_i)^2}{4\ln 2\alpha^2}} \tag{4}$$

Where $\alpha$ is the full width at the half maximum of the Gaussian and $A$ is a normalization factor. From Eqs. 1 and 3, we can derive the spectra of reduced resolution from original GOSAT and OCO-2 observations by

$$\mathbf{I}_{obs}^{\mathrm{deg}}(\lambda_i) = (\mathbf{H}_i * \mathbf{g} * \mathbf{I}_{rad})(\lambda_i) \tag{5}$$

$$= \mathbf{g} * \mathbf{I}_{obs}(\lambda_i) \tag{6}$$

The corresponding error covariance $\mathbf{S}_y^{\mathrm{deg}}$, which describes the measurement uncertainties of the target spectrometer, can be deduced from the original error covariance matrix $\mathbf{S}_y$ by

$$\mathbf{S}_y^{\mathrm{deg}} = \mathbf{g}\mathbf{S}_y\mathbf{g}^T . \tag{7}$$

Obviously, the degraded spectra need to be sampled according to the spectrometer's sampling ratio. For the MSR spectral sizing points in Table 1, the sampling ratios are 3.1, 3.1 and 3.3 at the $0.76~\mu\mathrm{m}$, $1.61~\mu\mathrm{m}$ and $2.06~\mu\mathrm{m}$ bands, respectively. This approach allows us to degrade high spectral resolution measurements to lower resolution measurements using the specification of the target instrument with the exception of the noise performance, which is adapted from the original GOSAT or OCO-2 spectrometer. Similarly, the forward model employs the same convolution in Eq. 5 before comparing the simulation with the degraded spectra. Thus both the satellite measurements and the forward model simulation as part of the retrieval are adapted accordingly.

Figure 1 shows an example of ISRF and spectra of OCO-2 in the $2.06~\mu\mathrm{m}$ band degraded to a spectral resolution of $0.55$ nm using a Gaussian $\mathbf{g}$ with a FWHM of $\alpha = 0.530$ nm. Analogously, we generated spectra with a resolution of $0.10$ and $0.30$ nm in the two other spectral bands as listed in Table 1 with $\alpha = 0.093$ and $0.294$, respectively.

With these modifications, we aim to evaluate the spectral sizing of ESA's concept for a $CO_2$ monitoring mission ($CO_2$M). In this study, we investigate retrieval performance of MSR type instrument under spectral resolutions of 0.097, 0.15, 0.30 and 0.55 nm for the $2.06~\mu\mathrm{m}$ band while recently a spectral resolution of 0.35 nm was specified for the $CO_2$M mission. It should be noted that, for a real MSR type instrument, the signal-to-noise (SNR) will be much higher than that of a degraded GOSAT or OCO-2 spectrum. Another limitation of using OCO-2 measurements, apart from adapting its SNR, is that the generated MSR type measurements are limited to the instrument's spectral range, which differs from the $CO_2$M mission. Retrieval results here are therefore not expected to be representative for the $CO_2$M mission adopting an MSR sizing approach.

## 3 Data

For our study, we considered OCO-2 observations only over land in the period from September 2014 to October 2017, which are spatio-temporally collocated within $3 \times 3$ degrees longitude-latitude and within 2 hours with $XCO_2$ ground-based observations of the TCCON network. Here, we use OCO-2 version 8 L1b data and obtained about $463,000$ soundings collocated with 16 TCCON sites as shown in Table 4. Analogously, we proceeded with GOSAT land observations (L1b version V201) for the years 2009-2016 using only 'high-gain' measurements of the instrument. Given the sparse spatial sampling of GOSAT, we employed a coarse spatial collocation criteria within 5 degrees latitude-longitude which results in $270,000$ individual observations collocated with observations from 10 different TCCON stations. Some TCCON sites are not used in this study mainly due to following two reasons:(1) limited overpass, for example, for high latitude sites and island sites. At high latitude area, satellite observations over land usually have low SNR and low Sun which has to be filtered out; (2) sites located within polluted or elevated areas, such as Caltech, USA and Zugspitze, Germany. As part of the processing chain, the data were filtered further with respect to latitudinal position, impact from regional $CO_2$ sources and terrain roughness. For both data sets,

we retrieved the column densities of $CO_2$ and in the case of GOSAT also $CH_4$ using the RemoTeC algorithm for measurements at their original resolutions. Subsequently, we reduced the spectral resolution to that of the MSR spectral sizing point of Table 1 assuming a fixed sampling ratio, as described in the previous section, and repeated the retrieval. To better understand the impact of the spectral resolution on $CO_2$ retrieval quality, the different MSR spectral sizing points included first a spectral degradation of the 0.76 $\mu$m band and 1.6 $\mu$m band of the original OCO-2 data to a resolution of 0.1 and 0.3 nm, respectively (MSR-a), and subsequently we gradually degraded the spectral resolution in the 2.06 $\mu$m band to 0.15, 0.30 and 0.55 nm while retaining the spectral resolutions in the 0.76 $\mu$m band and 1.6 $\mu$m band (MSR-b, MSR-c, MSR-d).

In order not to be affected by unknown instrument related issues such as spectrometer stray light, we generated simulated spectra for a global ensemble as described by Butz et al. (2009). The ensemble comprises 11,036 spectra and is designed to estimate retrieval errors induced by aerosol and cirrus for four typical days representing four seasons (Butz et al., 2012). In the ensemble, the description of aerosol and cirrus is much more complex than in the retrieval and so the assessment of the induced $XCO_2$ retrieval error can be used to estimate the scattering induced error for different spectral resolutions of the measurement. More details on the ensemble can be found in Butz et al. (2009, 2012); Hu et al. (2016).

## 4   Results

To start off our analysis, we would like to emphasize that in this work no bias correction is applied to the data. The differences between the $XCO_2$ retrieval product and the TCCON observations are summarized per station by the mean bias $b$ and the corresponding single sounding accuracy $\sigma$ defined by the root-mean-square deviation. To estimate the retrieval error caused by measurement noise, we use the mean of retrieval noise, which is obtained through linear error propagation in the retrieval. Additionally, we estimate the station-to-station variability $\sigma_s$ as the standard deviation of the mean biases among all TCCON sites to estimate the data product accuracy on regional scales, which is crucial for regional flux inversion. The validation with TCCON measurements is limited by its spatial coverage. To compensate the spatial sparseness of TCCON sites, we start with synthetic retrievals for global ensembles.

### 4.1   OCO-2 synthetic spectra

First, we studied the $XCO_2$ retrieval error for synthetic spectra calculated for the OCO-2 spectral ranges and resolutions and for the MSR-d type spectra derived from simulated OCO-2 measurements according to Section 2. The reported $XCO_2$ retrieval error is induced by the limited aerosol information that can be inferred from the measurement and the different sensitivity to the assumed measurement noise, which is on the level of the OCO-2 instrument (Mandrake et al., 2015). Any systematic error due to e.g. erroneous molecular spectroscopy or instrument calibration errors is not addressed here.

For performance evaluation, we considered the global ensemble as described in Section 3 without cirrus contamination and performed three different retrieval analyses:

**test-1** No radiometric offsets in the measurements.

**test-2** The OCO-2 radiance offsets identified by Wu et al. (2018) of $0.15\%$, $0.5\%$ and $0.14\%$ of the mean radiance of each band is added to the 0.76, 1.6 and 2.06 $\mu$m bands respectively. No radiometric offset is fit.

**test-3** Same radiometric offset as above but including a radiometric offset fit.

Table 3 shows the bias, single sounding accuracy and mean retrieval noise of synthetic OCO-2 and MSR-d measurements for the three test cases. We included all converged cases in our analysis without applying extra quality filtering. For test-1, aerosols induced a scatter in the retrieved $XCO_2$ with a single sounding accuracy of 2.7 and 3.1 ppm for OCO-2 and MSR-d synthetic measurements, respectively. Albeit with different sampling ratios, the mean retrieval noises are quite similar between OCO-2 and MSR-d synthetic measurements. When adding intensity offsets but not accounting for the offset in the retrieval (test-2), the OCO-2 and MSR-d retrievals exhibit similar single sounding accuracy as in test-1 but with an increased negative bias of $-2.70$ and $-2.30$ ppm, respectively. The results of test-3 indicate that for simulated measurements the radiometric offset can be fully mitigated by fitting a radiometric offset in each band as additional elements of the state vector for both OCO-2 and MSR-d measurements. However, we can not prove this for MSR-d type measurements reproduced from real OCO-2 observations. Moreover, test-1 and test-2 have similar noise-propagated errors but decreased single sounding precision in the case of moderate spectral sizing. For the $CO_2M$ mission, this will be partly mitigated by an MSR type instrument with an improved SNR performance.

Figure 2 and 3 show the global $XCO_2$ retrieval errors from the MSR-d and OCO-2 synthetic spectra for the test-1. In both cases, $XCO_2$ retrieval errors are typically smaller than 4 ppm in most regions. As discussed by Butz et al. (2012), aerosol introduced uncertainties strongly depend on the concentration, the profile and the micro-physical properties of the aerosol, like size distribution and refractive index, as well as on the surface albedo. Although it is difficult to identify the exact source of retrieval errors, we see that with reduced spectral resolution MSR-d retrievals have similar error distribution and global coverage as that of OCO-2. Large errors usually occur at high latitude regions with low surface albedo or in the Sahara and Asia with seasonal high aerosol loading.

## 4.2 OCO-2 TCCON validation

Due to the spatial sampling approach of the OCO-2 instrument with a continuous sampling in flight direction and with eight cross-track samplings, we typically obtain several collocations of OCO-2 measurements with TCCON observations for our collocation criteria. To evaluate the data quality, we consider overpass-averages both for the OCO-2 and TCCON $XCO_2$ data. This averaging helps to reduce the impact of random and representation errors in our comparison, where we assume that the latter shows a pseudo-random error pattern.

For OCO-2 around $386,600$ of the retrievals converged and $313,500$ finally passed the a posteriori quality filtering and are classified as 'good' quality data. Here, the overall data yield is similar to that reported by Wu et al. (2018). The OCO-2 retrievals have a global bias of $-2.50$ ppm, an averaged single sounding precision of $\sigma_a = 1.36$ ppm, a mean retrieval noise of 0.25 ppm and a station-to-station variability of $\sigma_s = 0.56$ ppm.

We first degraded the spectral resolution of the 0.76 $\mu$m band and 1.61 $\mu$m band but used the original measurements of 2.06 $\mu$m band (MSR-a). Subsequently, we gradually degraded the spectral resolution in the 2.06 $\mu$m band as described for the spectral sizing points MSR-b, MSR-c and MSR-d. We applied the same RemoTeC algorithm settings and similar quality filtering options as above. The filtering is adjusted to guarantee that the percentage of good quality retrievals in all four MSR type retrievals are around 67% as for the original OCO-2 data, although the number of overpasses per station can still differ for the different spectral sizing points.

Figure 4 summarizes $XCO_2$ retrieval performance for the MSR-d sizing point with an average single precision accuracy of $\sigma = 1.68$ ppm, a retrieval noise error of 0.83 ppm and a station-to-station variability of $\sigma_s = 0.56$ ppm. Here, the $XCO_2$ data product has a large negative global bias of $-6.97$ ppm, which is subtracted in the plot. The variation of biases between 16 different stations is depicted in Fig. 5 while the station-to-station variability $\sigma_s$ is more-or-less the same as OCO-2 retrievals.

To better understand these results and in particular the increase of the global bias, Table 5 summarizes the $XCO_2$ retrieval performance for OCO-2 and all MSR type measurements, i.e. also the MSR-a, MSR-b and MSR-c spectral sizing points. Here the overall data yield is very similar for the different data sets although differences may occur due to different percentage of convergence. Therefore, we also analyzed the results for the subset of identical data points, shown in Table 6.

From MSR-a type retrievals, we see that degrading the 0.76 $\mu$m band and 1.61 $\mu$m band has limited impact on the $XCO_2$ retrieval performance. For both selection approaches, lowering the spectral resolution in the 2.06 $\mu$m band causes an increase in single sounding precision, mean retrieval noise and mean bias, where the station-to-station variability shows little sensitivity to the different resolutions. Part of the scatter error can be attributed to retrieval noise, which is also gradually increased when lowering the spectral resolution. This part of the uncertainty will be reduced by an instrument with better SNR, which is the advantage of the MSR-type instruments.

The discrepancy in the mean bias could be for a large part due to intensity offset in the 2.06 $\mu$m band of OCO-2. As shown in Tables 5 and 6, the global mean bias increases greatly only when we degrade the 2.06 $\mu$m band. As reported by Wu et al. (2018), fitting additive intensity offsets to the two $CO_2$ absorption bands can improve both the accuracy and the single sounding precision of the $XCO_2$ retrieval. The fitted intensity offsets are also highly correlated ($r > 0.70$) with the mean signal in each band. This may hint at a stray light related radiometric error. Not fitting such an intensity offset reduces the depth of telluric absorption lines with respect to the continuum and so leads to an underestimation of the $CO_2$ column. The sensitivity to this radiometric error seems higher for low resolution spectra.

## 4.3 OCO-2 hot spot and regional gradient detection

One of the main objectives of the European $CO_2$ monitoring mission is to capture $CO_2$ variations from regional to local scales. In this section, we evaluate to what extent this capability is affected by a reduced spectral resolution of the MSR-c spectral sizing concept. To this end we use OCO-2 observations from the period 8 September to 31 October, 2014, and compare the OCO-2 and the spectrally degraded MSR-c retrievals over Europe, the Middle East and Africa and for two individual orbits with $XCO_2$ hot spots as presented by Nassar et al. (2017).

Figure 6 shows the OCO-2 and MSR-c $XCO_2$ product over the EMEA region, which include in total around $330,000$ individual data points. Here, we corrected both data sets with the corresponding mean bias of -2.50 and -6.03 ppm from Table 5. The OCO-2 and MSR-c retrievals in this region are highly correlated with $r = 0.80$, and the difference between corresponding cases have a standard deviation of 1.23 ppm. The two data sets have very similar $XCO_2$ distributions and both can well capture regional variations. For example, the low values of $XCO_2$ in East Europe of about 393 ppm and its increase in the Middle East to 396 ppm is clearly present in both data sets. Moreover, both $XCO_2$ products show enhancements to about 398 ppm towards Southern Africa due to seasonal biomass burning.

Nassar et al. (2017) reported on the OCO-2 capability to detect local $XCO_2$ emissions from coal power plants. Here we investigate to what extent this capability is affected by the spectral degradation of the MSR-c spectral sizing point. Figure 7 shows two orbits with $XCO_2$ emission plumes from the Sasan power plant in India and Ghent Generation station in Kentucky, US, as captured by OCO-2 and MSR-c type measurements. In both cases, the $XCO_2$ enhancement around power plants can be well captured by both the original OCO-2 and the MSR-c spectral sizing. Plume emissions depend on the $XCO_2$ enhancement with respect to background. In OCO-2 retrievals, the $XCO_2$ enhancements are about 7 ppm and 5 ppm around the Sasan and Ghent station, respectively. Compared to OCO-2 retrievals, MSR-c retrievals indicate an increased $XCO_2$ enhancement of about 1.5 ppm for both plume events. Since the estimated emission depends linearly on $XCO_2$ enhancement, the estimate of the spectrally degraded measurements of the MSR-c concept is about 20% to 30% higher than that from OCO-2 retrievals.

An important property of satellite observations in the shortwave infrared spectral range is the sensitivity to the total amount of $CO_2$ including the tropospheric boundary layer, which provides the key information to characterize $CO_2$ sources and sinks. The column averaging kernel describes this sensitivity showing the derivative of the retrieved $XCO_2$ with respect to changes in the $CO_2$ subcolumns as a function of height. It depends on the measurement error covariance, the regularization strength and the Jacobian matrix and is discussed in more detail by Butz et al. (2012). Figure 8 compares the averaging kernels for the different instrument concepts and shows that for all resolutions the retrieved $XCO_2$ product shows a stronger $CO_2$ sensitivity in the troposphere than in the stratosphere. Here the MSR-c retrievals have an increasing sensitivity down to the surface but a reduced sensitivity to stratospheric $CO_2$ while for OCO-2 the sensitivity stays more or less constant near the ground. This could be due to the fact that we have reduced sensitivity to pressure-dependent line-broadening effects under coarse spectral resolutions since we do not resolve individual $CO_2$ lines.

## 4.4 Study using GOSAT spectra

Finally, to compare our findings with independent GOSAT retrievals, we use, analogously to Galli et al. (2014), 270,000 GOSAT-TCCON collocations, where about $250,000$ successful retrievals pass the a posteriori quality filtering and are classified as 'good' quality retrievals. Although methane columns are retrieved simultaneously as in previous studies, we will focus here on the $XCO_2$ retrievals only. The difference with TCCON measurements at 10 sites shows an overall mean bias of $b = -2.28$ ppm, a single sounding accuracy of $\sigma_a = 2.01$ ppm, a mean retrieval noise of 0.62 ppm and a station-to-station variability of $\sigma_s = 0.42$ ppm. Compared with OCO-2 retrievals, GOSAT retrievals have similar mean bias but increased scatter and retrieval noise which is probably due to a higher noise level.

Using the approach of section 2.2, we convert GOSAT measurements to MSR-d measurements and repeat the full-physics retrieval and quality filtering. Figure 9 summarizes the MSR $XCO_2$ retrieval quality and number of observations per station. Almost the same number of observations converge and pass the quality filtering as for the original GOSAT retrievals. Figure 10 shows the variation of the bias and standard deviation among all 10 TCCON stations. Compared to the GOSAT retrievals, the global bias of the MSR retrieval decreases by 0.31 ppm while the station-to-station variability values increase slightly by 0.10 ppm. The mean retrieval noise increased to 1.22 ppm which is not shown in the figure. The reduced spectral resolution affects mainly the single sounding precision of $XCO_2$, which rises on average by 0.86 ppm and is exhibited at all TCCON stations. This is in agreement with the finding by Galli et al. (2014) and with the results from simulated measurements.

The increase in the scatter of the errors for low resolution spectra was already found for the simulated measurement ensemble and is in agreement with the OCO-2 findings of Section 4.2. In contrast to the OCO-2 analysis, we see for GOSAT data that the lower resolution has only a minor impact on the global mean bias. In turn, this suggests that the origin of this bias is not due to the interference of molecular spectroscopy but is most likely due to an OCO-2 specific feature, which did not occur in the corresponding GOSAT analysis. This can be attributed to the fact that GOSAT spectra benefit from TANSO-FTS's distinguishing features such as common field stop for all spectral bands thus can minimize stray light influence (Kuze et al., 2009).

## 5  Conclusions and discussion

We investigated the impact of spectral resolution on $XCO_2$ retrieval accuracy with current on-orbit satellite observations and synthetic measurements. From the study with GOSAT, OCO-2 and synthetic measurements, we conclude that the lower resolution of 0.1, 0.3 and 0.3-0.55 nm in the 0.76, 1.61 and 2.06 $\mu$m spectral bands mainly induces a larger scatter in the $XCO_2$ retrieval error, where the scatter gradually increases with lower spectral resolution. Part of the scatter error increase can be attributed to measurement noise, which can be reduced by MSR-type instruments with improved SNR. Both for GOSAT and OCO-2 measurements, the station-to-station variability is largely insensitive to a coarser spectral resolution. For GOSAT, the global $XCO_2$ bias differs little for the different spectral resolutions. This is not the case for OCO-2 measurements, which show a significant increase in the mean bias for decreasing spectral resolution. Most likely this increase is due to instrument related errors such as a radiance offset in the different bands. The investigation using OCO-2 and GOSAT observations are limited by the spatial spareness of TCCON sites. Therefore, we also investigate the impact of spectral resolution with synthetic spectra of global ensembles. The synthetic study confirms that single sounding precision decreases for low resolution and MSR type retrievals have similar systematic errors as OCO-2 for global ensembles. Finally, it should be noted that large part of uncertainty in $XCO_2$ retrievals from OCO-2, GOSAT or synthetic measurements still comes from pseudo-noise contribution of aerosols.

The $XCO_2$ enhancements due to localized hot spot emissions can be well captured by both spectral sizing concepts, the original OCO-2 measurements and the spectrally degraded measurements with about 20-30% difference in the estimated emission rate, as demonstrated for two $XCO_2$ plume events. Moreover, we found that the regional variation of $XCO_2$ in OCO-2 obser-

vations over Europe, Middle East and Africa is observed by both concepts with similar quality, where data of both retrievals were highly correlated with a correlation coefficient of 0.8 and a standard deviation of the differences of 1.23 ppm.

Currently, the European Commission (EC) and the European Space Agency (ESA) are considering a Copernicus $CO_2$ Monitoring system for monitoring anthropogenic $CO_2$ emissions using a spectrometer with moderate spectral resolution similar to the assumptions made in this study (Sierk et al., 2018). Aided by a dedicated multi-angle polarimeter (MAP), the system aims at providing $XCO_2$ products with a spatial resolution of 4 km$^2$ (over a > 200 km swath) with a single sounding accuracy better than 0.7 ppm and a systematic error less than 0.5 ppm. From our study, we see that the reduced resolution of OCO-2 and GOSAT measurements mainly reduce $XCO_2$ precision and have little effect on the station-to-station variability (the systematic error). Since a substantial contribution of the $XCO_2$ error from OCO-2, GOSAT and synthetic measurements comes from insufficient knowledge about the atmospheric light path, the $XCO_2$ retrieval accuracy will benefit from the measurements of the MAP aerosol instrument, which will well characterize aerosol contributions in the $CO_2$ absorption bands. The multi-angle polarimeter provides valuable information on aerosol micro-physical properties and aerosol height which exceeds the aerosol information that can be retrieved from the 3-band spectrometer such as GOSAT and OCO-2 (Mishchenko and Travis, 1997; Waquet et al., 2009; Dubovik et al., 2011; Hasekamp et al., 2011; Wu et al., 2016). Moreover, the increased scatter of the $XCO_2$ data will be mitigated by the targeted higher SNR performance of the $CO_2$ spectrometer.

This study is focused on the effect of a reduced spectral resolution on retrieval precision and accuracy using OCO-2 and GOSAT observation. It supports the spectral sizing of the future Copernicus mission but can not address the effects of enhanced SNR and broader spectral range in the 2.06 $\mu$m band, as targeted by the future $CO_2$ monitoring system. The study focus on the use of OCO-2 data with its specific radiometric performance, which thus do not fully cover the spectral range of the $CO_2$M mission. SNR requirements for the Copernicus candidate mission have been derived to meet the targeted single-sounding precision, taking into account the selected spectral resolution (Sierk et al., 2018).

*Data availability.* The OCO-2 L1b data (version 8) were provided by the OCO-2 project from the data archive at the NASA Goddard Earth Science Data and Information Services Center (https://daac.gsfc.nasa.gov/). TCCON data were obtained from the TCCON Data Archive (https://tccondata.org/). The MSR type retrieval results presented in this paper can be found at ftp://ftp.sron.nl/open-access-data/.

*Competing interests.* The authors declare that there is no conflict of interest.

*Acknowledgements.* This study was conducted in the context of the Spectral Sizing project funded by the European Space Agency (ESA) under contract no. ESA-IPL-PEO-FF-gp-LE-2016-456. The views expressed here can in no way be taken to reflect the official opinion of ESA.

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

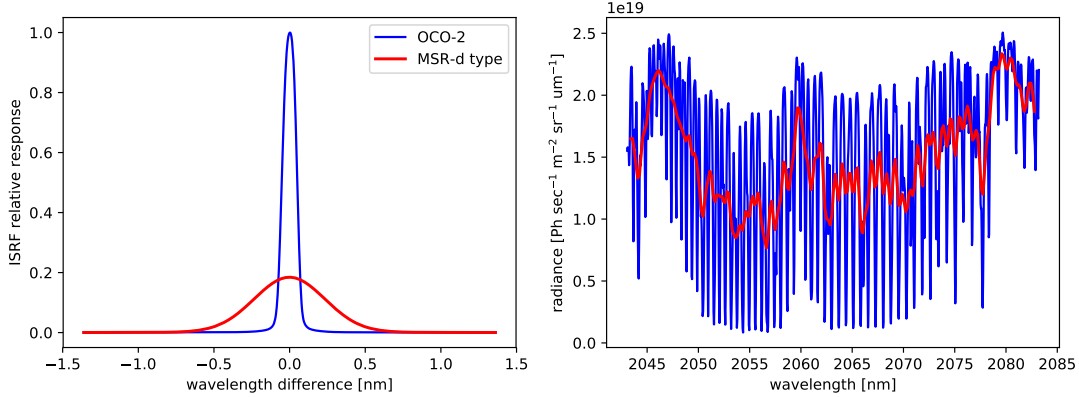

**Figure 1.** Example spectra and instrument spectral response functions of OCO-2 and MSR-d type instrument in the 2.06 $\mu$m band. Both OCO-2 and MSR-d ISRFs are scaled to the maximum of OCO-2 ISRF.

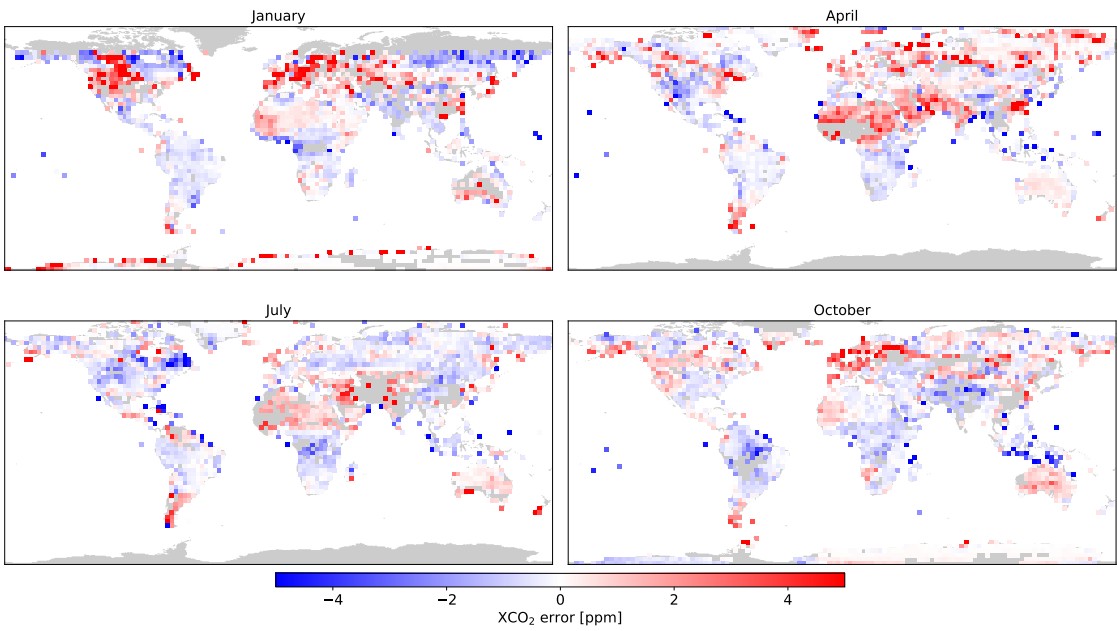

**Figure 2.** XCO$_2$ retrieval errors from MSR-d synthetic spectra of the test-1 for the global ensemble of Butz et al. (2009). Gray areas over land are not processed or retrievals do not converge.

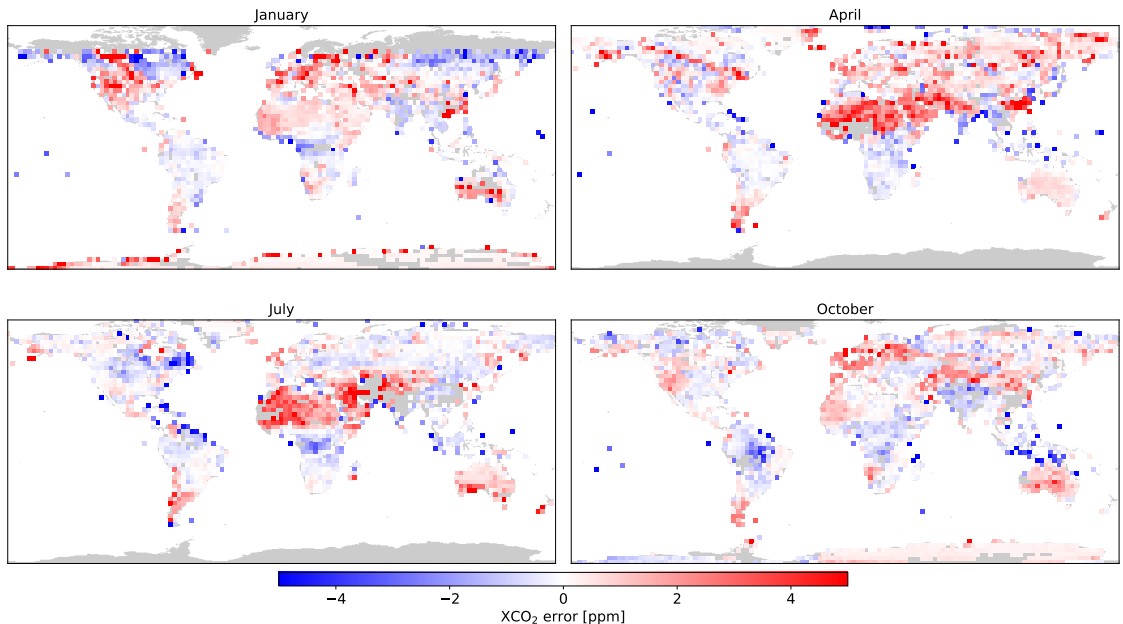

**Figure 3.** Similar as Fig. 3 but for OCO-2 type synthetic spectra.

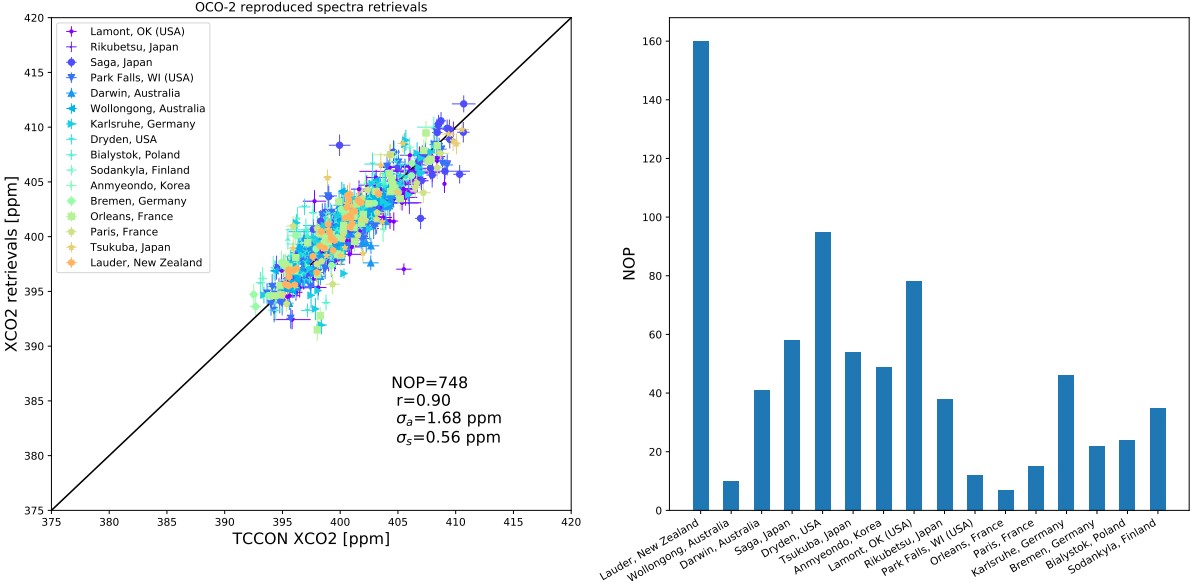

**Figure 4.** XCO$_2$ retrievals from MSR-d type spectra reproduced from OCO-2 measurements. The left panel shows the overall validation and the right panel shows the number of observations (NOBS) per station. In the left panel we included the total number of observations (n), overall bias (b), single sounding accuracy ($\sigma$), station-to-station variability ($\sigma_s$), Pearson correlation coefficient (r) and the one-to-one line. We subtracted a global bias of $b = -6.97$ ppm.

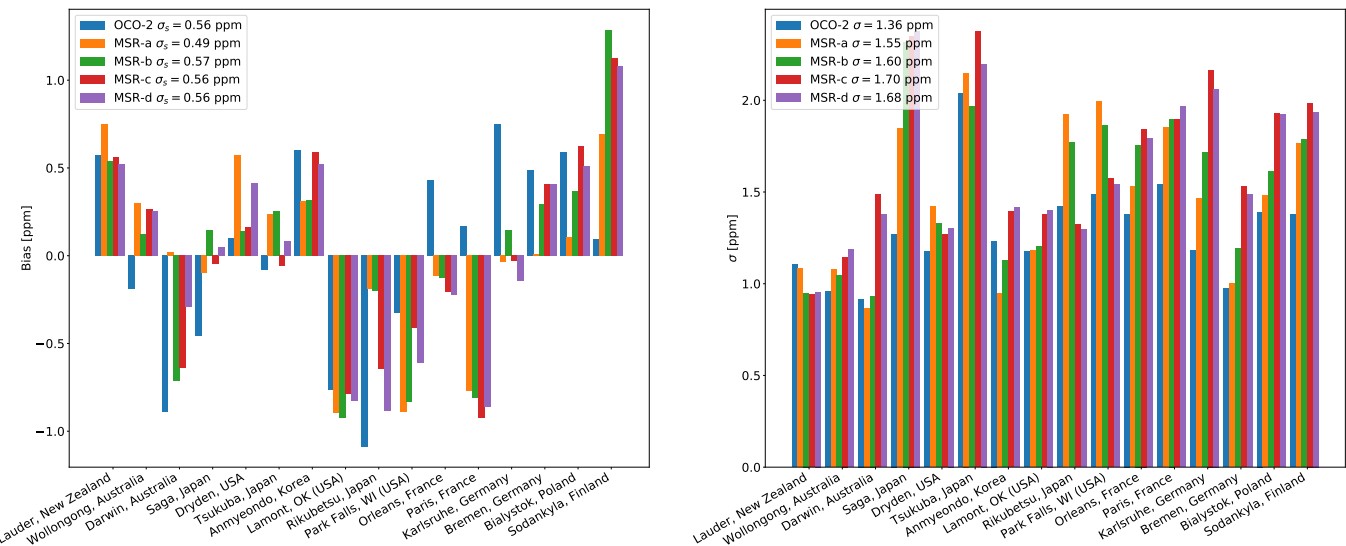

**Figure 5.** Bias and standard deviation ($\sigma$) at different TCCON stations for OCO-2 and MSR type retrievals. Mean biases are subtracted accordingly for OCO-2 and MSR type retrievals to show the bias variation on the same reference level. The station-to-station variability ($\sigma_s$) and single sounding accuracy ($\sigma$) is included in the left and right panel legends, respectively. Here, MSR type measurements are reproduced from OCO-2 measurements.

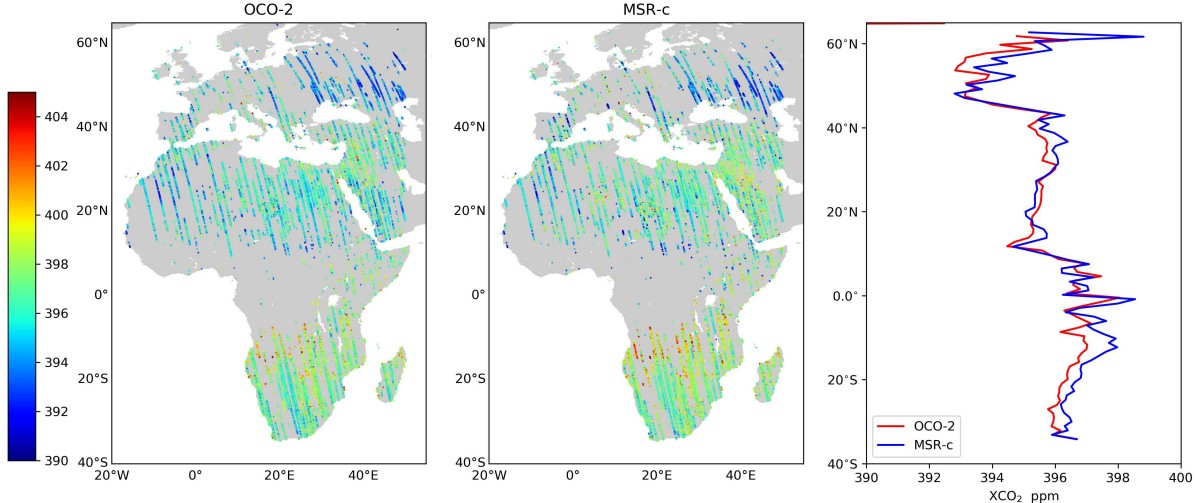

**Figure 6.** XCO$_2$ retrievals over Europe, the Middle East and Africa regions using OCO-2 and MSR-c type measurements. We processed all orbits obtained by OCO-2 between 8 September and 31 October 2014. For each type of retrieval, the corresponding mean bias in Table 5 is subtracted. In the right most panel, we include latitude variation of XCO$_2$ averaged over longitude.

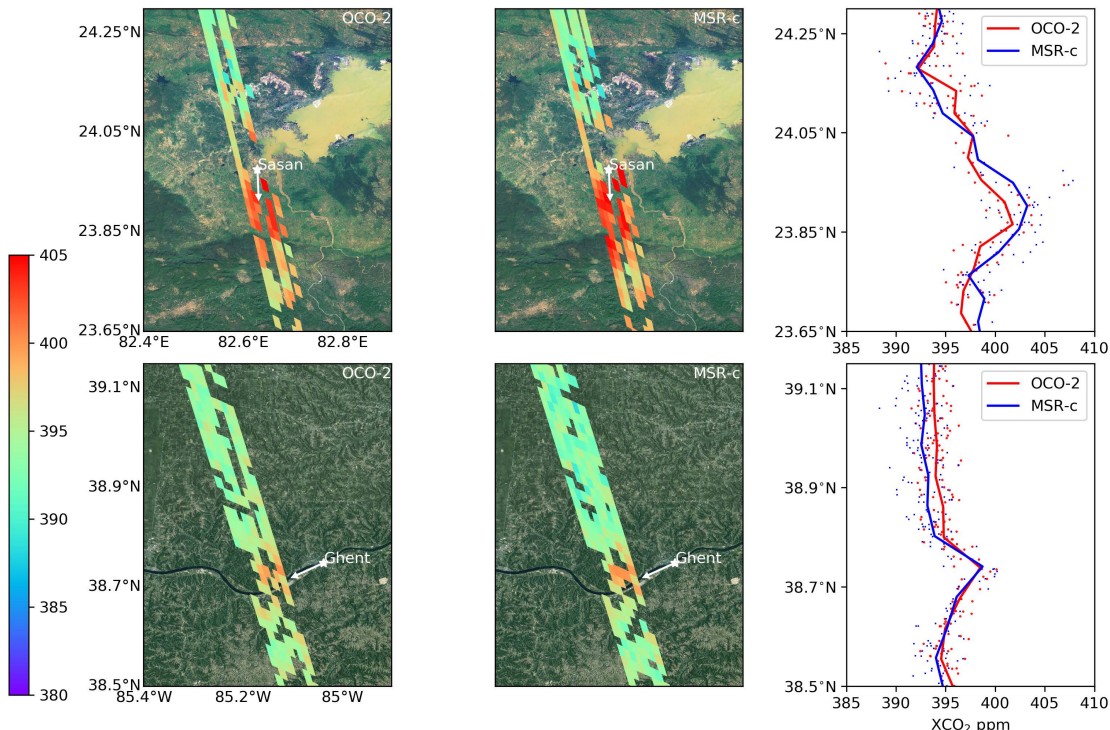

**Figure 7.** $XCO_2$ retrievals along orbits with hot spots as observed by OCO-2 and MSR-c type instruments. Local potential sources (power plant) are marked by asterisks and the directions of the local wind are marked with arrows. For each hotspot overpass, $XCO_2$ values (scatter dots) and median values (solid lines) along the orbit are shown in the right most panel. Here, OCO-2 passes by the Sasan and Ghent power plants on Octorber 23, 2014 and August 13, 2015, respectively. Mean biases reported in Table 5 are removed from each orbit.

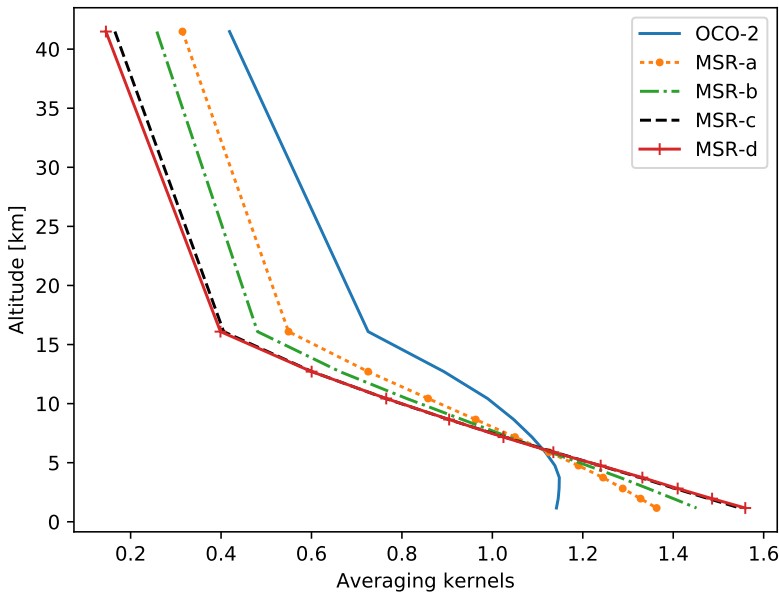

**Figure 8.** Example of an averaging kernel of OCO-2 and MSR type retrievals in RemoTeC. The observation is obtained close to the TCCON site Lamont under nadir mode with a solar zenith angle of 20.5 degrees. Averaging kernels are plotted as a function of central height of the $CO_2$ atmosphere sub-column.

**Table 1.** Spectral resolutions of the OCO-2 and GOSAT instruments and the four spectral sizing points MSR-a to MSR-d with reduced spectral resolution, which are investigated in this study. Here the spectral sizing point of MSR-d is adapted from the CarbonSat design. The list signal-to-noise ratios per spectral sampling for each instrument concept are caculated under the same incoming radiance of 75.2, 10.4 and 2.4 W/m2/sr/$\mu$m for the 0.76, 1.61 and 2.06 $\mu$m band, respectively (Sierk et al., 2018).

| | Spectral ranges [nm] | Resolution[nm]/Sampling ratio | | | Signal-to-noise ratio at the reference radiance |
| --- | --- | --- | --- | --- | --- |
| | | 0.76 $\mu$m | 1.61 $\mu$m band | 2.06 $\mu$m band | |
| OCO-2 | 758-772, 1591-1621, 2042-2081 | 0.042/2.5 | 0.076/2.5 | 0.097/2.5 | 426, 964, 497 |
| MSR-a | 747-773, 1590-1675, 1925-2095 | 0.1/3.1 | 0.3/3.1 | 0.097/2.5 | 590, 1720, 497 |
| MSR-b | 747-773, 1590-1675, 1925-2095 | 0.1/3.1 | 0.3/3.1 | 0.15/3.3 | 590, 1720, 538 |
| MSR-c | 747-773, 1590-1675, 1925-2095 | 0.1/3.1 | 0.3/3.1 | 0.30/3.3 | 590, 1720, 760 |
| MSR-d | 747-773, 1590-1675, 1925-2095 | 0.1/3.1 | 0.3/3.1 | 0.55/3.3 | 590, 1720, 1030 |
| GOSAT | 758-775, 1560-1720, 1920-2080 | 0.015/1.4 | 0.08/2.7 | 0.1/2.7 | 340, 952, 486 |

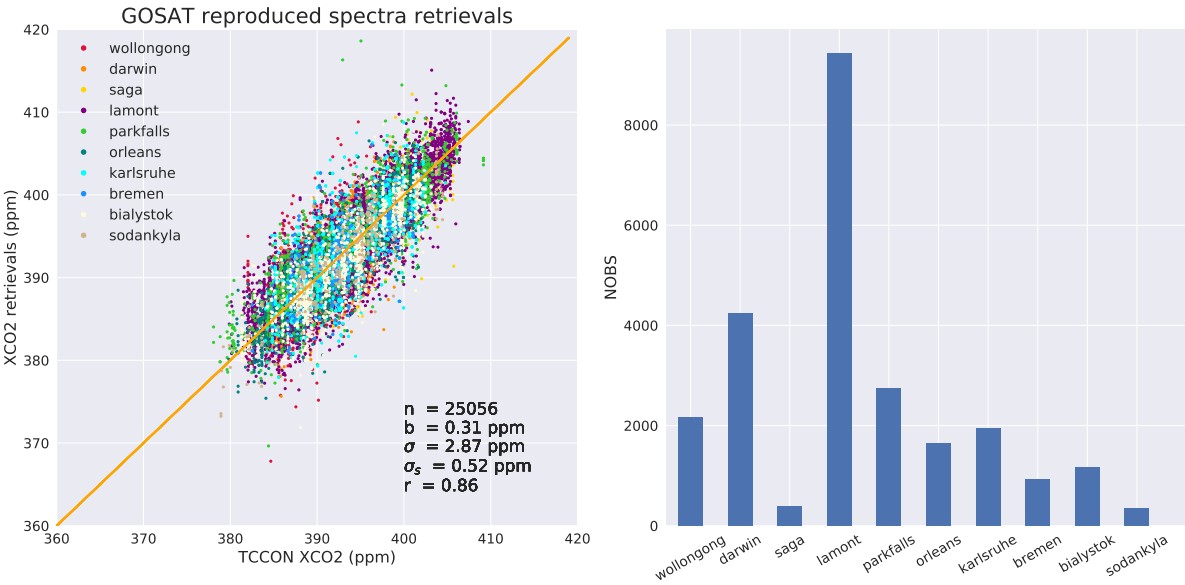

**Figure 9.** XCO$_2$ retrievals from MSR-d type spectra reproduced from GOSAT measurements. As in Fig. 4, we included the statistical diagnostics of the study.

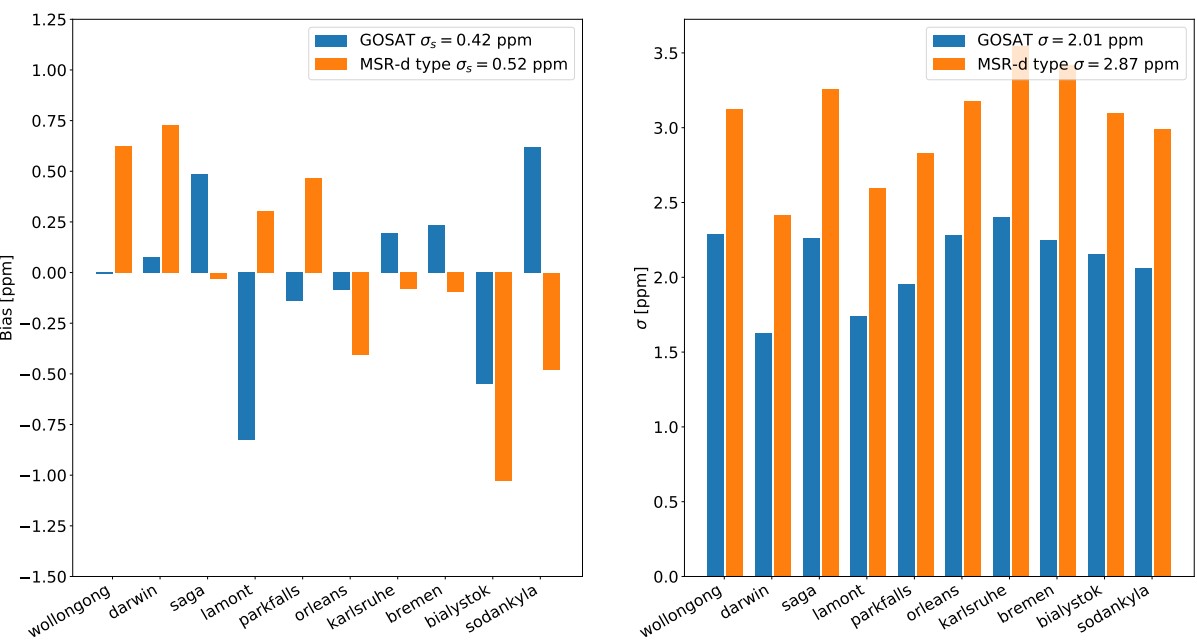

**Figure 10.** Similar as Fig. 5, bias and standard deviation ($\sigma$) at different TCCON stations for GOSAT and MSR-d type retrievals. Here, MSR-d type measurements are reproduced from GOSAT measurements.

**Table 2.** Settings of the filters used for excluding low-quality XCO$_2$ retrievals in OCO-2 retrievals.

| Parameter | Definition | Allowed Range |
|---|---|---|
| sza | Solar zenith angle | val$\leq 70°$ |
| vza | Viewing zenith angle | val$\leq 45°$ |
| iter | Number of retrieval iterations | val$\leq 30$ |
| dfs | Degrees of Freedom for Signal for CO$_2$ | val$\geq 1.0$ |
| $\chi^2$ | Overall goodness of fit | val$\leq 10.0$ |
| $\chi^2_{1st}$ | Goodness of fit in O$_2$ A-band | val$\leq 20.0$ |
| SNR1 | Signal noise ratio in the 0.76 $\mu$m band | val$\geq 100$ |
| SNR3 | Signal noise ratio in the 2.06 $\mu$m band | val$\geq 100$ |
| Blended albedo* | 2.4×albedo_NIR - 1.13×albedo_SWIR-2 | val$\leq 1.0$ |
| sev | Surface elevation variation | val$\leq 75$ m |
| $\alpha_s$ | Aerosol size parameter | $3.0 \leq$val$\leq 10.0$ |
| $\tau_{0.765}$ | Aerosol optical depth in O$_2$ A-band | val$\leq 0.35$ |
| Aerosol ratio parameter | $\tau_{0.765}*z_s/\alpha_s$, $z_s$ is aerosol layer height | val$\leq 300$ m |
| Xerr | Retrieval uncertainty for X$_{CO_2}$ | val$\leq 2.0$ ppm |
| Ioff$_1$ | Fitted Intensity offset ratio in the 0.76 $\mu$m band | $-0.005 \leq$val$\leq 0.015$ |

*The blended albedo filter was first introduced in Wunch et al. (2011).

**Table 3.** XCO$_2$ retrieval performance for synthetic OCO-2 and MSR-d measurements. Intensity offsets are added to spectra in test-2 and test-3 but only fitted for test-3. The bias and the single sounding accuracy are the mean and standard deviation of differences between retrievals and truths, respectively. Noise errors are retrieval uncertainties from linear noise propagation.

| | Bias [ppm] | | Single sounding accuracy [ppm] | | mean retrieval noise [ppm] | | Convergence percentage | |
|---|---|---|---|---|---|---|---|---|
| | OCO-2 syn | MSR-d syn | OCO-2 syn | MSR-d syn | OCO-2 syn | MSR-d syn | OCO-2 syn | MSR-d syn |
| test-1 | 0.04 | 0.05 | 2.69 | 3.10 | 0.60 | 0.57 | 81% | 71% |
| test-2 | -2.70 | -2.30 | 2.83 | 2.97 | 0.59 | 0.58 | 82% | 77% |
| test-3 | -0.01 | -0.44 | 2.10 | 1.97 | 1.01 | 1.20 | 69% | 66% |

**Table 4.** List of TCCON stations used in the study

| Stations | Latitude and Longitude | Reference |
|---|---|---|
| Sodankyla, Finland | (67.3N, 26.6E) | Kivi et al. (2014) |
| Bialystok, Poland | (53.2N, 23.0E) | Deutscher et al. (2015) |
| Bremen, Germany | (53.1N, 8.8E) | Notholt et al. (2014) |
| Karlsruhe, Germany | (49.1N, 8.4E) | Hase et al. (2015) |
| Park Falls, WI(USA) | (48.4N, 2.3E) | Wennberg et al. (2014) |
| Paris, France | (48.4N, 2.3E) | Te et al. (2014) |
| Orleans, France | (47.9N, 2.1E) | Warneke et al. (2014) |
| Rikubetsu, Japan | (43.4N, 143.7E) | Morino et al. (2016b) |
| Lamont, OK(USA) | (36.6N, 97.4W) | Wennberg et al. (2016) |
| Anmyeondo, Korea | (36.5N, 126.3E) | Goo et al. (2014) |
| Tsukuba, Japan | (36.0N, 140.1E) | Morino et al. (2016a) |
| Dryden, USA | (34.9N, 117.8W) | Iraci et al. (2016) |
| Saga, Japan | (33.2N, 130.2E) | Kawakami et al. (2014) |
| Darwin, Australia | (12.4S, 130.9E) | Griffith et al. (2014a) |
| Wollongong, Australia | (34.4S, 150.8E) | Griffith et al. (2014b) |
| Lauder, New Zealand | (45.0S, 169.6E) | Sherlock et al. (2014) |

**Table 5.** $XCO_2$ retrieval performance for OCO-2, MSR-a, MSR-b, MSR-c and MSR-d type measurements under similar throughput. Here, MSR type measurements are generated using OCO-2 measurements.

| | Resolution [nm] | bias | $\sigma_a$[ppm] | $\sigma_s$ [ppm] | mean retrieval noise [ppm] | Overpass | Single sounding accuracy [ppm] |
|---|---|---|---|---|---|---|---|
| OCO-2 | 0.042, 0.076, 0.097 | -2.50 | 1.37 | 0.56 | 0.25 | 783 | 2.14 |
| MSR-a | 0.1, 0.3, 0.076 | -1.46 | 1.55 | 0.49 | 0.42 | 782 | 2.16 |
| MSR-b | 0.1, 0.3, 0.15 | -3.79 | 1.60 | 0.57 | 0.46 | 778 | 2.29 |
| MSR-c | 0.1, 0.3, 0.30 | -6.03 | 1.70 | 0.55 | 0.54 | 745 | 2.26 |
| MSR-d | 0.1, 0.3, 0.55 | -6.97 | 1.68 | 0.56 | 0.80 | 748 | 2.31 |

**Table 6.** Same as Table 5, but for the intersection between OCO-2 and MSR type retrievals.

| | Resolution [nm] | bias | $\sigma_a$ [ppm] | $\sigma_s$ [ppm] | mean retrieval noise [ppm] | Overpass | Single sounding accuracy [ppm] |
|---|---|---|---|---|---|---|---|
| OCO-2 | 0.042, 0.076, 0.097 | -2.00 | 1.33 | 0.55 | 0.25 | 669 | 2.05 |
| MSR-a | 0.1, 0.3, 0.097 | -1.17 | 1.39 | 0.46 | 0.39 | 669 | 2.08 |
| MSR-b | 0.1, 0.3, 0.15 | -3.52 | 1.47 | 0.54 | 0.44 | 669 | 2.23 |
| MSR-c | 0.1, 0.3, 0.30 | -5.73 | 1.55 | 0.59 | 0.59 | 669 | 2.34 |
| MSR-d | 0.1, 0.3, 0.55 | -6.73 | 1.58 | 0.59 | 0.83 | 669 | 2.41 |