# Peer review of "XCO2 observations using satellite measurements with moderate spectral resolution: Investigation using GOSAT and OCO-2 measurements"

_Atmospheric Measurement Techniques, 2019_

## Short Comment (SC1) · 19 Sep 2019

Dear authors,

this manuscript uses TCCON data from several stations but lacks the proper references to the TCCON dataset DOIs. Citing these is a requirement of the TCCON Data Use Policy. Luckily, this can still be fixed easily.

Instructions:

- Go to the TCCON data archive at https://tccondata.org/.

- Visit the pages of the TCCON stations that you used for your study.

- Look at the section "Cite this record as:". You can download the dataset DOI citation in various formats - including the correct one for Copernicus Publications.

- These dataset DOI references should be included in your manuscript's list of references.

- Since you used data from several TCCON stations, I recommend adding a table that contains all the TCCON site names, locations and the citations (like "Wennberg et al., 2016" for Lamont) . This is easier than putting all the citations into the text.

- Note: some TCCON stations have several data releases (R1 etc.). Every release has a different DOI, so please make sure that you cite the dataset release that you actually used in your study.

The DOI references are also included in the header of the TCCON netCDF files that you downloaded originally. However, the DOI metadata on the TCCON Data Archive is probably more up to date.

Citing the TCCON dataset DOIs is essential for tracking the use of TCCON data. Running a TCCON station requires considerable funds and efforts by highly qualified personnel. Being able to track the use of their data is critical for most TCCON stations to continue operations.

Additional comments:

- in your acknowledgments, the URL of the TCCON Data Archive is wrong. The correct one is https://tccondata.org/.

Please fix these issues before the final version. Thanks!

Kind regards

Dietrich Feist

---

## Referee Comment (RC1) · Anonymous Referee #1 · 23 Sep 2019

<General Comments>

Land-coverage for CO2 measurements with existing space-borne instruments is sparse. Wide swath measurement with moderate spectral resolutions is important for global flux estimations. Spatial resolutions higher than OCO-2 and GOSAT will improve estimations of local flux. This paper describes the optimization of spectral resolution using data acquired by OCO2- and GOSAT for much denser observations. Bias and standard deviation of the retrieved CO2 density are well described. The paper is worth publishing. However, I have the following general comments. Before publishing the paper, major revisions are needed.

[Figure]

(1) Page 2, Line 28 "unprecedent accuracy and precision" Line 30 "XCO2 precision <0.7ppm and systematic error <0.5ppm". These requirements are challenging. How to improve accuracy and precision of existing GOSAT and OCO-2 with lower spectral resolution should be described. It is not clear that SNR and precision of XCO2 retrieval are linearly correlated.

(2) Authors mention there are instrument specific errors in OCO-2 and GOSAT. They should be discussed in more detail. My understanding on GOSAT and OCO-2 instruments is as follows. The spectral quality with GOSAT FTS is good with symmetric instrument function and low stray light. However, SNR is lower than OCO-2. GOSAT FTS uses a single pixel detector mounted on its optical axis. Optical aberration is small and its instrument line shape function becomes symmetrical. GOSAT FTS has a common field stop for all bands and uses the modulated part of the interferogram. Theoretically stray light is low.

(3) Authors mention the largest error source is aerosol and proposes an auxiliary aerosol sensor. However, aerosol related errors are not well described. "Aerosols induce a scatter" (Page 8, Line 212): "pseud-noise contribution of aerosol" (Page 11, line 324): More detailed explanations are needed. Which parameters is critical, optical thickness, size, type, or height? Why does the MAP instrument reduce uncertainty?

<Specific Comments>

(1) Page 1, Abstract Brief description of the proposed spectral range, spatial resolution and coverage, and required SNR in the abstract will help readers' understanding.

(2) Page 4, Lines 240, 244-245 The main reason for global bias of 6.97ppm should be discussed in more detail. Lines 244-245 are difficult to understating Detailed explanation on "different algorithm convergence" is needed.

(3) Page 19, Figure 5, Averaging kernel All the MSRs have similar slopes in averaging kernels. AK is quite different from the one for OCO-2. What is the main reason for the

large difference between OCO-2 and MSRs? How do the authors estimate local flux quantitatively from XCO2 with sloped AK?

(4) Page 20, Figure 6 Observation dates should be specified.

(5) Page 21, Figure 7 Legends are not the same as those for Figure. 2 "$\sigma$= 2.87ppm" > "$\sigma$a= 2.87ppm" > Is "b" global bias? Is it already subtracted?

<Technical Corrections>

(1) Page 6, Line 150 Sy=gSygT > Sydeg=gSygT

(2) Page 22, Table 2, "$\sigma$s" What is the unit of "aerosol size parameter"?

(3) page 23, Table 5, "SD" Is it standard deviation?

---

## Referee Comment (RC2) · Anonymous Referee #3 · 22 Oct 2019

This manuscript analyses the impact of spectral resolution on the performance of a CO$_2$ satellite mission, which is relevant for future large-swath imaging instruments. To this end, satellite measurements from OCO-2 and GOSAT are spectrally degraded and the retrieval results are compared to corresponding results using the original spectral resolution. It is concluded from the spectrally degraded satellite measurements and from synthetic measurements that the lower resolution mainly induces a larger random error and has only little effect on the systematic error.

The manuscript covers an important topic, falls into the scope of AMT, and is well written. However, I think that a more detailed analysis is necessary to support the

conclusion that a degradation of the spectral resolution does not impact systematic errors significantly. Therefore, I recommend publication after major revisions have been incorporated.

**General Comments**

The main concern is the discussion of the systematic errors depending on the spectral resolution.

1) The validation with the TCCON shows that the station-to-station variability (standard deviation of the local biases) is similar for the retrievals based on the degraded and original spectral resolution. However, the local biases for the MSR and original instrument at a fixed site can differ considerably (see Figures 3 and 8). Hence, the good agreement of the standard deviations may become worse when adding or removing specific sites and is possibly not representative globally. This should be discussed in the manuscript. It would be helpful to harmonise and maximise the sites used in the OCO-2 and GOSAT comparison. Is it possible to add additional high latitude sites, e.g., East Trout Lake and Eureka?

2) Since the spatial representativity of the TCCON comparison is limited, the analysis of the synthetic spectra is particularly important to assess the impact of the spectral resolution on the systematic errors globally. Unfortunately, the corresponding discussion is rather short and the results are condensed to a single number "bias" in Table 3. How is this number defined? It would be very desirable to show the errors based on the global ensemble on seasonal global maps like in Butz et al. (2012) for OCO-2 and for all MSR concepts a-d to better track the impact of successive spectral degradation on the systematic error and to check if the decreased convergence rate clusters in certain regions.

See also specific comments for more details.

**Specific Comments**

*OCO-2 synthetic spectra*

More details are needed here. Please show and discuss the seasonal global maps of the errors obtained from the ensemble for the original OCO-2 spectral resolution and for MSR-d at least for test-1. If possible, it would be very helpful to also show maps for all MSR concepts a-d as proposed in the general comments. Moreover, it would be beneficial to show an additional map in each case for a 2-band retrieval without the 2.06 $\mu$m band (MSR-e) and to extent Table 3 accordingly to verify that the spectral resolution of 0.55 of MSR-d in this band is actually useful to reduce systematic errors.

*TCCON validation*

Please harmonise and maximise the sites used in the OCO-2 and GOSAT comparison, if possible. Additional high latitude sites would be particularly interesting. Please also harmonise the ordering of sites in Figure 2 (arbitrarily?), Figure 3 (by latitude), and Figures 7&8 (alphabetically); I would prefer to sort the sites by latitude in all Figures.

Please add and discuss bars to include all MSR concepts (a-d and ideally the proposed 2-band test MSR-e) in Figures 3 and 8 to better track the impact of successive spectral degradation.

*OCO-2 hot spot and regional gradient detection*

Why is MSR-c used in this section and not MSR-d as before?

Are the different averaging kernels considered in the comparisons of Figures 5 and 6? Is it possible that the 20-30% higher enhancement for the MSR-c concept in Figure 6 is due to the increased surface sensitivity of the spectrally degraded concept (see Figure 4)?

*Conclusions and discussion*

Please adjust the conclusions concerning systematic errors depending on the spectral resolution according to the new analyses or weaken the conclusions in terms of the

general comments above.

Is there a reference showing that the MAP instrument will actually characterise aerosol contributions in the $CO_2$ absorption bands well and that the $XCO_2$ retrieval accuracy "will benefit greatly" from its measurements? Otherwise, please weaken the conclusions by saying that the MAP instrument is aiming at reducing systematic errors.

**Technical Corrections**

P6, EQ7: Replace "$S_y =$" by "$S_y^{deg} =$"

P6, L153: 3.29 here, but 3.3 in Table 1

P8, L230: 1.37 ppm here, but 1.36 ppm in Figure 3

P23, Table 5: Replace "SD" by "$\sigma_a$"

**References**

Butz, A., Galli, A., Hasekamp, O., Landgraf, J., Tol, P., and Aben, I.: TROPOMI aboard Sentinel-5 Precursor: Prospective performance of $CH_4$ retrievals for aerosol and cirrus loaded atmospheres, Remote Sensing of Environment, 120, 267–276, 2012.

---

## Author Comment (AC1) · 18 Dec 2019

We thank all reviewers for their constructive comments, which helped to improve the paper. Below, we address all comments point-by-point.

**Referee #3,**

Dear authors,
this manuscript uses TCCON data from several stations but lacks the proper references to the TCCON dataset DOIs. Citing these is a requirement of the TCCON Data Use Policy. Luckily, this can still be fixed easily.
Instructions:
- Go to the TCCON data archive at https://tccondata.org/.
- Visit the pages of the TCCON stations that you used for your study.
- Look at the section "Cite this record as:". You can download the dataset DOI citation
in various formats - including the correct one for Copernicus Publications.
- These dataset DOI references should be included in your manuscript's list of references.
- Since you used data from several TCCON stations, I recommend adding a table that contains all the TCCON site names, locations and the citations (like "Wennberg et al., 2016" for Lamont) . This is easier than putting all the citations into the text.

- Note: some TCCON stations have several data releases (R1 etc.). Every release has a different DOI, so please make sure that you cite the dataset release that you actually used in your study. The DOI references are also included in the header of the TCCON netCDF files that you downloaded originally. However, the DOI metadata on the TCCON Data Archive is probably more up to date. Citing the TCCON dataset DOIs is essential for tracking the use of TCCON data. Running a TCCON station requires considerable funds and efforts by highly qualified personnel. Being able to track the use of their data is critical for most TCCON stations to continue operations.
Additional comments:
- in your acknowledgments, the URL of the TCCON Data Archive is wrong. The correct one is https://tccondata.org/.
Please fix these issues before the final version. Thanks!
Kind regards
Dietrich Feist

R1- Thanks for the suggestion. We now add a Table (see below) in the paper to cite all the TCCON references.

**Table 4.** List of TCCON stations used in the study

| Stations | Latitude and Longitude | Reference |
| --- | --- | --- |
| Sodankyla, Finland | (67.3N, 26.6E) | Kivi et al. (2014) |
| Bialystok, Poland | (53.2N, 23.0E) | Deutscher et al. (2015) |
| Bremen, Germany | (53.1N, 8.8E) | Notholt et al. (2014) |
| Karlsruhe, Germany | (49.1N, 8.4E) | Hase et al. (2015) |
| Park Falls, WI(USA) | (48.4N, 2.3E) | Wennberg et al. (2014) |
| Paris, France | (48.4N, 2.3E) | Te et al. (2014) |
| Orleans, France | (47.9N, 2.1E) | Warneke et al. (2014) |
| Rikubetsu, Japan | (43.4N, 143.7E) | Morino et al. (2016b) |
| Lamont, OK(USA) | (36.6N, 97.4W) | Wennberg et al. (2016) |
| Anmyeondo, Korea | (36.5N, 126.3E) | Goo et al. (2014) |
| Tsukuba, Japan | (36.0N, 140.1E) | Morino et al. (2016a) |
| Dryden, USA | (34.9N, 117.8W) | Iraci et al. (2016) |
| Saga, Japan | (33.2N, 130.2E) | Kawakami et al. (2014) |
| Darwin, Australia | (12.4S, 130.9E) | Griffith et al. (2014a) |
| Wollongong, Australia | (34.4S, 150.8E) | Griffith et al. (2014b) |
| Lauder, New Zealand | (45.0S, 169.6E) | Sherlock et al. (2014) |

---

## Author Comment (AC2) · 18 Dec 2019

We thank all reviewers for their constructive comments, which helped to improve the paper. Below, we address all comments point-by-point.

**Anonymous Referee #1,**

<General Comments>
Land-coverage for CO2 measurements with existing space-borne instruments is sparse. Wide swath measurement with moderate spectral resolutions is important for global flux estimations. Spatial resolutions higher than OCO-2 and GOSAT will improve estimations of local flux. This paper describes the optimization of spectral resolution using data acquired by OCO2- and GOSAT for much denser observations. Bias and standard deviation of the retrieved CO2 density are well described. The paper is worth publishing. However, I have the following general comments. Before publishing the paper, major revisions are needed.

(1) Page 2, Line 28 "unprecedent accuracy and precision" Line 30 "XCO2 precision <0.7ppm and systematic error <0.5ppm". These requirements are challenging. How to improve accuracy and precision of existing GOSAT and OCO-2 with lower spectral resolution should be described. It is not clear that SNR and precision of XCO2 retrieval are linearly correlated.
R1-Not changed: There are many design options under discussion to improve the radiometric performance of the instrument compared to a spectrometer such as OCO-2. The manuscript focuses on one particular, major design aspect, the choice of the spectral resolution. SNR and precision of XCO2 retrieval as defined in the manuscript are NOT linearly correlated. Large part of XCO2 retrieval uncertainties are due to lightpath modification by aerosol scattering. A possible mitigation is to include a multiangle polarimeter as we discussed the conclusions of the manuscript. In this paper, we focus on the investigation of the impact of moderate spectral resolution of the main spectrometer. The joint XCO2 retrievals using measurements of the spectrometer and MAP are currently under development and a manuscript in preparation.

(2) Authors mention there are instrument specific errors in OCO-2 and GOSAT. They should be discussed in more detail. My understanding on GOSAT and OCO-2 instruments is as follows. The spectral quality with GOSAT FTS is good with symmetric instrument function and low stray light. However, SNR is lower than OCO-2. GOSAT FTS uses a single pixel detector mounted on its optical axis. Optical aberration is small and its instrument line shape function becomes symmetrical. GOSAT FTS has a common field stop for all bands and uses the modulated part of the interferogram. Theoretically stray light is low.
R2-Thanks for the detailed suggestions. We have adjusted this part and added the sentence "This can be attributed to the fact that GOSAT spectra benefit from TANSO-FTS's distinguishing features such as common field stop for all spectral bands thus can minimize stray light influence." at Page 11 line 326. Moreover, at page 4 line 106, we added the sentence "Measurements of the OCO-2 push-bloom spectrometer with high SNR includes most likely larger stray light errors than the TANSO-FTS (Thermal And Near infrared Sensor for carbon Observation - Fourier Transform Spectrometer) on-board GOSAT.".

(3) Authors mention the largest error source is aerosol and proposes an auxiliary aerosol sensor. However, aerosol related errors are not well described. "Aerosols induce a scatter" (Page 8, Line 212): "pseud-noise contribution of aerosol" (Page 11, line 324): More detailed explanations are needed. Which parameters is critical, optical thickness, size, type, or height? Why does the MAP instrument reduce uncertainty?
R3-We added a new paragraph in the section 4.1 (page 8, line 230) to explain the synthetic retrieval results and aerosol introduced errors. "….. As discussed by Butz et al. (2012), aerosol introduced errors strongly depend on the concentration, the profile and the micro-physical properties of the aerosol like size distribution and refractive index as well as on surface albedo. Although it is difficult to identify the most

critical aerosol parameter as it is finally the effect on the light path which induces the error, we see that with reduced spectral resolution MSR-d retrievals yield a similar error distribution and a global data coverage as that of the OCO-2 data product."

Moreover, we added a sentence to the conclusions to explain why the MAP instrument can reduce the aerosol induced errors (Page 12, line 356), "The multi-angle polarimeter provides valuable information on aerosol micro-physical properties and aerosol height, which exceeds the aerosol information that can be retrieved from a 3-band spectrometer such as GOSAT and OCO-2 (Mishchenko and Travis, 1997; Waquet et al., 2009; Dubovik et al., 2011; Hasekamp et al., 2011; Wu et al., 2016)."

<Specific Comments>
(1) Page 1, Abstract Brief description of the proposed spectral range, spatial resolution and coverage, and required SNR in the abstract will help readers' understanding.
R4- Thanks for the suggestion. The abstract now includes (Page 1, line 5) "The future European CO2 monitoring constellation, currently undergoing feasibility studies at the European Space Agency (ESA), is targeting a moderate spectral resolution of 0.1, 0.3 and 0.3-0.55 nm in the three spectral bands with high signal-to-noise (SNR) ratio as well as a spatial resolution of 4 km$^2$ and a across-track swath width $>250$ km.".

(2) Page 4, Lines 240, 244-245 The main reason for global bias of 6.97ppm should be discussed in more detail. Lines 244-245 are difficult to understating Detailed explanation on "different algorithm convergence" is needed.
R5-The sentence is revised to "Here, the overall data yield is very similar for the different data sets although differences may occur due to different percentage of convergence." (Page 9, line 259). The large global bias is presented in MSR type retrievals and discussed with more details at both last paragraph of section 4.2 (Page 9, line 267) and last paragraph of section 4.4 (Page 11, line 323).

(3) Page 19, Figure 5, Averaging kernel All the MSRs have similar slopes in averaging kernels. AK is quite different from the one for OCO-2. What is the main reason for the large difference between OCO-2 and MSRs? How do the authors estimate local flux quantitatively from XCO2 with sloped AK?
R6-The main reason for the different shapes of the averaging kernel is that a spectrometer with a fine spectral resolution can resolve individual lines, which are pressure and temperature dependent. With a reduced spectral resolution, this sensitivity is reduced. We included the sentence on page 10, line 303 of our manuscript "This could be due to the fact that we have reduced sensitivity to pressure-dependent line-broadening effects under coarse spectral resolutions since we do not resolve individual CO2 lines." For a classical data assimilation, the averaging kernel can be considered in the flux inversion and so the shape differences should not cause a difficulty. In the paper, we do not estimate the effect of a different AK shape but estimate the sensitivity of the local flux to errors in XCO2, which we consider in the worst case to be proportional to the XCO2 bias.

(4) Page 20, Figure 6 Observation dates should be specified.
R7-We adapted the manuscript accordingly and the observation dates are now included in the Figure.

(5) Page 21, Figure 7 Legends are not the same as those for Figure. 2 "σ= 2.87ppm" > "σa= 2.87ppm" > Is "b" global bias? Is it already subtracted?
R8- Not changed: Figure 7 is XCO2 retrievals from GOSAT reproduced spectra while figure 2 is from OCO-2. Here, b is the overall mean bias without subtraction.

<Technical Corrections>

(1) Page 6, Line 150 Sy=gSygT > Sydeg=gSygT

(2) Page 22, Table 2, "σs" What is the unit of "aerosol size parameter"?

(3) page 23, Table 5, "SD" Is it standard deviation?

R9- The aerosol size parameter $\sigma_s$ is a unitless parameter. A sentence of  "Here, the size parameter $\alpha$ is unitless." is added at Page 4 line 98. The "SD" is replaced with "$\sigma_a$" as in Table 4.

---

## Author Comment (AC3) · 18 Dec 2019

We thank all reviewers for their constructive comments, which helped to improve the paper. Below, we address all comments point-by-point.

**Anonymous Referee #2,**

This manuscript analyses the impact of spectral resolution on the performance of a CO 2 satellite mission, which is relevant for future large-swath imaging instruments. To this end, satellite measurements from OCO-2 and GOSAT are spectrally degraded and the retrieval results are compared to corresponding results using the original spectral resolution. It is concluded from the spectrally degraded satellite measurements and from synthetic measurements that the lower resolution mainly induces a larger random error and has only little effect on the systematic error.

The manuscript covers an important topic, falls into the scope of AMT, and is well written. However, I think that a more detailed analysis is necessary to support the conclusion that a degradation of the spectral resolution does not impact systematic errors significantly. Therefore, I recommend publication after major revisions have been incorporated.

General Comments

The main concern is the discussion of the systematic errors depending on the spectral resolution.
1) The validation with the TCCON shows that the station-to-station variability (standard deviation of the local biases) is similar for the retrievals based on the degraded and original spectral resolution. However, the local biases for the MSR and original instrument at a fixed site can differ considerably (see Figures 3 and 8). Hence, the good agreement of the standard deviations may become worse when adding or removing specific sites and is possibly not representative globally. This should be discussed in the manuscript. It would be helpful to harmonise and maximise the sites used in the OCO-2 and GOSAT comparison. Is it possible to add additional high latitude sites, e.g., East Trout Lake and Eureka?
R1- Indeed, the limitation of spatial coverage of TCCON sites should be emphasized in the paper. Therefore, we added the sentences (Page 7, line 204) "The validation with TCCON measurements is limited by its spatial coverage. To compensate the spatial sparseness of TCCON sites, we start with synthetic retrievals for global ensembles."
    In the study, we included as many TCCON sites as possible. We used 16 sites in OCO-2 validation and 10 sites in GOSAT validation. Some sites are not used in this study due to following reasons:
(1) limited overpass, for example, for high latitude sites and island sites. At high latitude area, satellite observations over land usually have low SNR and low Sun which has large uncertainties and has to be filtered out.;
(2) sites located within polluted or elevated areas, such as Caltech, USA and Zugspitze, Germany.
    Explanations are now added at (Page 6, line 179) of "Some TCCON sites are not used in this study mainly due to following two reasons:(1) limited overpass, for example, for high latitude sites and island sites. At high latitude area, satellite observations over land usually have low SNR and low Sun which has to be filtered out; (2) sites located within polluted or elevated areas, such as Caltech, USA and Zugspitze, Germany.".
    In summary, we are afraid that it is not possible to include TCCON sites like East Trout Lake and Eureka into the validation since we do not have enough good quality retrievals around those high latitude stations. In the paper, the TCCON station Sodankyla Finland (67.3668N, 26.6310E) is the one at highest latitude and so represents high latitude regions.
    To compensate the limitation of the TCCON comparison, in the original manuscript we included the regional comparison between OCO-2 and MSR-c type retrievals in Section 4.3, although only relative differences can be studied in this manner.

2) Since the spatial representativity of the TCCON comparison is limited, the analysis of the synthetic spectra is particularly important to assess the impact of the spectral resolution on the systematic errors globally. Unfortunately, the corresponding discussion is rather short and the results are condensed to a single number "bias" in Table 3. How is this number defined? It would be very desirable to show the errors based on the global ensemble on seasonal global maps like in Butz et al. (2012) for OCO-2 and for all MSR concepts a-d to better track the impact of successive spectral degradation on the systematic error and to check if the decreased convergence rate clusters in certain regions.

R2-The bias is the mean of differences between XCO2 retrievals and truth of all the good quality retrievals. We add a sentence in the caption of Table 3 to explain the definition of the bias, "The bias and the single sounding accuracy are the mean and standard deviation of differences between retrievals and truths, respectively."

  For the global error distribution maps, we added Figure 2 and 3, which include global XCO2 retrievals errors for MSR-d and OCO-2 type measurements. A paragraph is added in Section 4.1 (Page 8, line 230) to describe the comparison,

> "Figure 2 and 3 show the global XCO2 retrieval errors from the MSR-d and OCO-2 synthetic spectra for the test-1. In both cases, XCO2 retrieval errors are typically smaller than 4 ppm in most regions. As discussed by Butz et al. (2012), aerosol introduced uncertainties strongly depend on the concentration, the profile and the micro-physical properties of the aerosol, like size distribution and refractive index, as well as on the surface albedo. Although it is difficult to identify the exact source of retrieval errors, we see that with reduced spectral resolution MSR-d retrievals have similar error distribution and global coverage as that of OCO-2. Large errors usually occur at high latitude regions with low surface albedo or in the Sahara and Asia with seasonal high aerosol loading.".

[Figure]

**Figure 2.** XCO$_2$ retrieval errors from MSR-d synthetic spectra in the test-1 for the global ensemble of Butz et al. (2009). Gray areas over land are not processed or retrievals do not converge.

[Figure]

**Figure 3.** Similar as Fig. 3 but for OCO-2 type synthetic spectra.

See also specific comments for more details.

OCO-2 synthetic spectra

More details are needed here. Please show and discuss the seasonal global maps of the errors obtained from the ensemble for the original OCO-2 spectral resolution and for MSR-d at least for test-1. If possible, it would be very helpful to also show maps for all MSR concepts a-d as proposed in the general comments. Moreover, it would be beneficial to show an additional map in each case for a 2-band retrieval without the 2.06 µm band (MSR-e) and to extent Table 3 accordingly to verify that the spectral resolution of 0.55 of MSR-d in this band is actually useful to reduce systematic errors.

R3-Seasonal global maps of the errors are now included in Figure 2 and Figure 3 for MSR-d and OCO-2 type spectra, respectively.

The 1.6 um band is normally used for the so-called proxy retrieval (Schepers et al., 2012). For the CO2M mission, the baseline is that the 2.06 um band is available. Thus, the 2-band retrieval (NIR+1.6 um) is not discussed in the paper. However, we made a preliminary study using only the NIR and 1.6 um band for XCO2 full-physics retrieval. For the 2-band retrieval, we use the same retrieval settings for aerosol properties as that in 3-band retrievals. From the comparison shown in Fig. S1, we see that retrieval errors increase when omitting the 2.0 µm band. The standard deviation of the errors increases by more than 1.0 ppm when compared with that of 3-band retrievals. We are convinced that 2.06 um band is important for characterizing aerosol properties in the retrieval.

[Figure]

Figure S1. XCO2 retrieval error as a function of aerosol optical depth in the NIR band. The 2-band and 3-band retrieval for MSR-d concept are shown in left and right panels, respectively. The overall bias is similar in both retrievals while the standard deviation of errors is increased by more than 1.0 ppm in 2-band retrievals.

TCCON validation

Please harmonise and maximise the sites used in the OCO-2 and GOSAT comparison, if possible. Additional high latitude sites would be particularly interesting. Please also harmonise the ordering of sites in Figure 2 (arbitrarily?), Figure 3 (by latitude), and Figures 7&8 (alphabetically); I would prefer to sort the sites by latitude in all Figures. Please add and discuss bars to include all MSR concepts (a-d and ideally the proposed 2-band test MSR-e) in Figures 3 and 8 to better track the impact of successive spectral degradation.

R4- Thanks for the suggestion. We have adjusted those figures accordingly. Figure 2 and Figure 7 and 8 are now sorted by latitude. Figure 3 (Figure 5 in the revised version) is updated to include the variation of biases for all MSR concepts. Figure 8 already includes both GOSAT and MSR-d type retrievals.

[Figure]

**Figure 5.** Bias and standard deviation ($\sigma$) at different TCCON stations for OCO-2 and MSR type retrievals. Mean biases are subtracted accordingly for OCO-2 and MSR type retrievals to show the bias variation on the same reference level. The station-to-station variability ($\sigma_s$) and single sounding accuracy ($\sigma$) is included in the left and right panel legends, respectively. Here, MSR type measurements are reproduced from OCO-2 measurements.

OCO-2 hot spot and regional gradient detection

Why is MSR-c used in this section and not MSR-d as before? Are the different averaging kernels considered in the comparisons of Figures 5 and 6? Is it possible that the 20-30% higher enhancement for the MSR-c concept in Figure 6 is due to the increased surface sensitivity of the spectrally degraded concept (see Figure 4)?

R5- MSR-c is used because at current stage the CO2M mission is considering a spectral resolution around 0.3 nm in the 2.06 um band. In the comparisons of Figures 5 and 6, averaging kernels have been taken into account and that differences in the averaging kernel leads to difference null-space contribution in the product. It is difficult to conclude if the 20-30 % difference in the enhancements come from different prior contribution or from different retrieval biases due to different spectral resolution.

Conclusions and discussion
Please adjust the conclusions concerning systematic errors depending on the spectral resolution according to the new analyses or weaken the conclusions in terms of the general comments above.

R6-The conclusions are adjusted following reviewer's comments. In the conclusion section, we added the sentences (Page 11, line 338) "The investigation using OCO-2 and GOSAT observations are limited by the spatial sparseness of TCCON sites. Therefore, we also investigated the impact of spectral resolution with synthetic spectra of global ensembles."

Is there a reference showing that the MAP instrument will actually characterise aerosol contributions in the CO2 absorption bands well and that the XCO2 retrieval accuracy "will benefit greatly" from its measurements? Otherwise, please weaken the conclusions by saying that the MAP instrument is aiming at reducing systematic errors.

R7- The synergistic use of a MAP and a XCO2 spectrometer is a new research field, a corresponding study is ongoing and a manuscript in preparation. Therefore, we have weakened the conclusions in the paper.

Technical Corrections
P6, EQ7: Replace "S y =" by "S y deg ="
P6, L153: 3.29 here, but 3.3 in Table 1
P8, L230: 1.37 ppm here, but 1.36 ppm in Figure 3
P23, Table 5: Replace "SD" by "σ a "

R9-Thanks for the corrections. These is now adjusted accordingly in the paper.

References
Butz, A., Galli, A., Hasekamp, O., Landgraf, J., Tol, P., and Aben, I.: TROPOMI aboard Sentinel-5 Precursor: Prospective performance of CH 4 retrievals for aerosol and cirrus loaded atmospheres, Remote Sensing of Environment, 120, 267–276, 2012.
Schepers, D. and Guerlet, S. and Butz, A. and Landgraf, J. and Frankenberg, C. and Hasekamp, O. and Blavier, J. F. and Deutscher, N. M. and Griffith, D. W. T. and Hase, F. and Kyro, E. and Morino, I. and Sherlock, V. and Sussmann, R. and Aben, I. (2012) Methane retrievals from Greenhouse Gases Observing Satellite (GOSAT) shortwave infrared measurements: Performance comparison of proxy and physics retrieval algorithms. Journal of Geophysical Research. Atmospheres, 117 (D10).